# Hemodynamic Effects of Subaortic Stenosis on Blood Flow Characteristics of a Mechanical Heart Valve Based on OpenFOAM Simulation

**DOI:** 10.3390/bioengineering10030312

**Published:** 2023-03-01

**Authors:** Aolin Chen, Adi Azriff Basri, Norzian Bin Ismail, Kamarul Arifin Ahmad

**Affiliations:** 1Department of Mechanical Engineering, Faculty of Engineering, University Putra Malaysia, Serdang 43400, Selangor, Malaysia; chenaolin0729@gmail.com; 2Department of Aerospace Engineering, Faculty of Engineering, University Putra Malaysia, Serdang 43400, Selangor, Malaysia; adiazriff@upm.edu.my; 3Department of Medicine, Faculty of Medicine and Health Sciences, University Putra Malaysia, Serdang 43400, Selangor, Malaysia; norziani@upm.edu.my

**Keywords:** subaortic stenosis, mechanical heart valve, CFD, heart valve diseases

## Abstract

Subaortic stenosis (SAS) is a common congenital heart disease that can cause significant morbidity and mortality if not treated promptly. Patients with heart valve disease are prone to complications after replacement surgery, and the existence of SAS can accelerates disease progression, so timely diagnosis and treatment are required. However, the effects of subaortic stenosis on mechanical heart valves (MHV) are unknown. This study aimed to investigate flow characteristics in the presence of subaortic stenosis and computationally quantify the effects on the hemodynamics of MHV. Through the numerical simulation method, the flow characteristics and related parameters in the presence of SAS can be more intuitively observed. Based on its structure, there are three types of SAS: Tunnel-type SAS (TSS); Fibromuscular annulus SAS (FSS); Discrete SAS (DSS). The first numerical simulation study on different types of SAS found that there are obvious differences among them. Among them, the tunnel-type SAS formed a separated vortex structure on the tunnel-type narrow surface, which exhibits higher wall shear force at a low obstacle percentage. However, discrete SAS showed obvious differences when there was a high percentage of obstacles, forming high peak flow, high wall shear stress, and a high-intensity complex vortex. The presence of all three types of SAS results in the formation of high-velocity jets and complex vortices in front of the MHV, leading to increased shear stress and stagnation time. These hemodynamic changes significantly increase the risk of MHV dysfunction and the development of complications. Despite differences between the three types of SAS, the resultant effects on MHV hemodynamics are consistent. Therefore, early surgical intervention is warranted in SAS patients with implanted MHV.

## 1. Introduction

Subaortic stenosis is a heart valve disease that is characterized by a narrowing of the area below the aortic valve. This narrowing can obstruct the flow of blood from the left ventricle to the aorta [1]. SAS can occur alone or in combination. It usually presents with obstruction, left ventricular hypertrophy, aortic regurgitation (AR) [2], and rapid growth accompanied by high-velocity jets and high-pressure gradients on the LVOT.

SAS usually occurs at the fibrous membrane, the muscular crest, or a combination of both [3]. Depending on the structure, SAS can be divided into three types. The first and most common type is discrete subaortic stenosis; it is often described as a thin membrane, but it usually takes the form of an indistinguishable fibrous muscle layer 1–2 mm thick [4]. The second type is a thicker fibrous muscle ridge, most commonly described as a fibromuscular annulus. Lastly, tunnel-type SAS usually presents as long, narrow fibromuscular tracts [5]. The latter two are also collectively referred to as “fixed subaortic stenosis.” SAS caused by thinner fibrous membranes is more focal. Conversely, fibromuscular ridges cause more diffuse obstruction, often resulting in tunnel-type lesions associated with more significant stenosis [6].

SAS can lead to a range of complications, including valve inflammation, scarring, and eventual apical thickening. Furthermore, SAS may arise as part of an obstructive lesion complex that typically includes mitral stenosis and coarctation of the aorta [5]. The peak systolic phase jet flow impacts the aortic valve flaps, leading to harm, scarring, excess flaps, and prolapse, making the valve more vulnerable to malfunction and damage [7]. Prosthetic valve replacement surgery on the aortic valve is needed in cases of aortic stenosis or insufficiency. A mechanical heart valve (MHV) is an excellent option because it is durable [8]. However, it is more likely to lead to complications such as thrombosis, so patients must take anticoagulants for life [9].

Early detection of valve dysfunction and stenotic obstruction is the key to successful treatment. Surgery is the only effective treatment for SAS, and postoperative recurrence is common, occurring in 8–34% of patients, making it challenging to decide whether and when to intervene [10]. Doppler hemodynamic parameters are used to evaluate the need for the procedure. By continuously measuring the jet velocity with the Doppler beam well aligned with the flow axis, the pressure gradient (*ΔP*) and Doppler velocity index (DVI) can be estimated [11,12]. The combination of transthoracic and transesophageal echocardiography with Doppler imaging is commonly employed to detect and evaluate the degree of severity in cases of SAS. Focal or diffuse left ventricular outflow tract stenosis is estimated by instantaneous peak systolic differential pressures of >20 mmHg. Peak transient gradients of >80 mm Hg are considered severe [13].

Computational fluid dynamics (CFD) strategies are used to study blood flow properties in the presence of SAS. Subaortic stenosis produces a high-velocity jet and a mean transvalvular pressure gradient (TMPG), and LVOT systolic blood flow disorder forms rich and complex vortex dynamics [3]. The presence of fluid wall shear stress (WSS) abnormalities in the LV anatomy of the SAS has also been found [14]. Computational models have been used to assess SAS and reveal changes in leaflet motion due to obstruction; however, in these models, the simulation of natural AV function and leaflet deformation remains challenging [1]. Previous LV aortic valve flow simulations have addressed this challenge by specifying the applied leaflet displacement as boundary conditions or by modeling the valve as a static or rigid structure [15]. Currently, there is a lack of research that has examined the effects of disruptions or disorders in the left ventricular outflow tract (LVOT) on hemodynamics and function [14]. Utilizing patient-specific computational fluid dynamics modeling can increase our understanding of SAS (Guivier et al.). Additionally, the evaluation of subaortic stenosis due to hypertrophic cardiomyopathy shows that occlusion alters leaflet kinematics [16]. Therefore, simplifying LVOT flow conditions severely affects valve hemodynamics and leaflet WSS [17]. Massé DD et al. verified that the abnormal geometry of LVOT can generate an abnormal shear force on the septal wall and then trigger a fibrotic reaction [10].

Although CFD has been used in the study of SAS, its application is mainly focused on studies about DSS. Due to the different structures, there are many types of SAS, such as discrete subaortic stenosis and fixed subaortic stenosis. The differences between the different types of SAS have not been fully studied. To bridge this knowledge gap, this study aimed to computationally quantify the flow and wall shear characteristics of different types of SAS at different percentages of blockage. The models included three different types of SAS and four different degrees of obstruction. The results of this study may pave the way for a deeper investigation into the possible implications of the pathogenesis of SAS. The impact of subaortic stenosis on MHV manifestations was also explored in this study.

## 2. Methods

This section illustrates the development of the mitral heart valve (MHV) model. Furthermore, the numerical approach, boundary conditions, and governing equations are examined.

### 2.1. Geometry

This research utilized the 23-mm On-X heart valve. The model was generated using SolidWorks software, with structural information obtained from Mirkhani et al. [18]. The simulation domain consisted of four components: the front aortic section, the MHV, the Valsalva sinus, and the back aortic section, as illustrated in Figure 1. Depending on the structural differences, there were three separate types of SAS: TSS (tunnel SAS), FSS (fibromuscular annulus SAS), and DSS (discrete SAS).

TSS: tunnel-type SAS (long, narrow fibromuscular tracts).

FSS: fibromuscular annulus SAS (thicker fibromuscular ridges).

DSS: discrete SAS (described as 1–2 mm of semilunar valve-like fibromuscular membrane composition [19]).

In each case, there were four groups with different blocking percentages of 10%, 20%, 30%, and 40%, depending on the obstruction height, as shown in Figure 2. TSS had a length of 28 mm and a smooth entrance at a 60° angle on the left. FSS had a base 10 mm wide. In patients without aortic insufficiency, the mean membrane-valve distance was 6 mm [20], so the center of the DSS and the end of the tunnel SAS were set to be 6 mm away from the MHV leaflet. For the DSS, the fibromuscular base width was 1.5 mm.

### 2.2. Boundary Conditions

The simulation was carried out under pulsatile conditions, utilizing an experimental pulsatile flow as the inlet condition, and applying typical physical flow conditions of approximately 25 L/min peak flow rate and a 0.86 s cardiac cycle [13]. The current study concentrates on the forward macroscale flow characteristics downstream of the MHV, with one-third of the cardiac cycle dedicated to systole, as shown in Figure 3. The present study is focused on investigating the hemodynamic behavior of the mechanical heart valve during the full opening phase from 45 to 280 ms, and thus it does not account for leaflet opening and closing phases. Specifically, the study aims to investigate the interaction between blood flow and valve leaflets during the peak flow phase, while also analyzing the formation of complex eddies and turbulence and the distribution of shear stress on the valve surface, where 
$T_{1}\;$
 = 0.09 s at the mid-acceleration phase and 
$T_{2}\;$
 = 0.19 s at peak systole.

### 2.3. Governing Equation

This study utilized the open-source toolbox OpenFOAM and the pimpleFoam unsteady incompressible solver [21]. Blood was assumed to be an incompressible fluid, with the equation of continuity and momentum as follows [22]:
(1)
∇·u=0


(2)
$$\frac{\partial \left(\rho u\right)}{\partial t}+\nabla \cdot \left(\rho uu\right)\;\nabla \cdot \left(\mu \nabla u\right)-\left(\nabla u\right)\cdot \nabla \mu =-\nabla p$$

where *u* is the velocity vector, *p* is the pressure, *μ* is the viscosity, and *ρ* is the density. The density value was set at 1060 kg/m^3^ [23]. The governing equation is based on the OpenFOAM solver [23]. The pimpleFoam solvers in OpenFOAM can automatically adjust the time step, with a time step of 
$1e^{-6}$
) and maxCo set to 1 to limit the timestep adjustment. To simulate the shear-thinning behavior of blood, the Navier-Stokes equations were integrated with an appropriate non-Newtonian constitutive model of dynamic viscosity; in this case, the well-known Carreau model was used [24]. 
$\eta_{0}$
 is the viscosity at zero shear rate, 
$\eta_{\infty }$
 is the viscosity at infinite shear rate, *λ* is a time constant of the fluid, 
$\dot{\gamma }\;$
is the shear rate, and *n* is a dimensionless parameter that characterizes the degree of shear-thinning.

(3)
$$\eta \left(\dot{\gamma }\right)=\eta_{\infty }+\left(\eta_{0}-\eta_{\infty }\right){\left[1+{\left(\lambda \dot{\gamma }\right)}^{2}\right]}^{\frac{n-1}{2}}$$

where: 
$\eta_{0}=0.56\;Pa\;s,\;\;\eta_{\infty }=0.0035\;Pa\;s,$

λ=8.2, n=0.2128
.

### 2.4. Validation

In this study, the cfMesh tools were utilized to generate the mesh. cfMesh is a library for mesh generation that is implemented within the OpenFOAM framework. The meshDict can be divided into mandatory settings (surfaceFile and maxCellsize) and local refinement (boundaryCellsize and localRefinement). The overall mesh size is set through the mandatory settings, and the localRefinement function encrypts the aortic and leaflet walls, as shown in Figure 4.

The grid independence was verified using five groups of different grid numbers and by comparing the maximum wall shear stress (WSS) obtained, as illustrated in Figure 5. When the number of grids reached approximately three million, the WSS stabilized, and the number of grids did not increase significantly. The final mesh count was determined to be 3.02 million, in which the minimum cell size was set at 1 mm, the maximum cell size was set at 5 mm, for refinement, the aortic wall was set at 2 mm, and the leaflet wall was set at 0.7 mm to ensure the accuracy of mesh generation. In addition, the current CFD model was compared with the numerical and experimental results of Bluestein et al. [25] by the blood flow of normal BMHV around the systolic peak. Velocity profiles were acquired downstream of the valve and in proximity to the caudal region of the leaflet and were normalized to the plotted velocity and location. As depicted in Figure 6, the results of this study were found to be in good agreement with previous studies.

## 3. Results

### 3.1. Velocity Contours

In the systole phase, the velocity profile of the entire field was investigated under normal cardiac output at two different instants: 100 ms (mid-acceleration phase) and 200 ms (peak systole phase). The structure with MHV and different percentages of SAS (10%, 20%, 30%, and 40%) are shown in Figure 7. In the absence of SAS, the blood flow through the MHV experienced acceleration and the formation of three-jet flow (one central and two sides), followed by dissipation into a chaotic flow state. When blood flows through the SAS, the existence of SAS triggers the acceleration of blood flow in advance, and the degree of acceleration depends on the percentage and shape of SAS. In the absence of SAS or in cases of low SAS percentages, a distinct triple jet pattern is observed, which interacts as the three jets flow and generates a secondary flow pattern that dissipates the jet. As the percentage of the SAS barrier increases, the interaction between the three jets intensifies, causing the blood flow to advance towards dissipation. The impact of SAS types, specifically FSS and DSS, on early dissipation is pronounced, with DSS having the most significant impact on accelerating blood flow under the same obstacle percentage.

The peak velocity value was proportional to the severity of SAS, as shown in Figure 8a. Different types of subaortic stenosis result in varying degrees of increase in peak velocity. The peak velocity of the FSS (at 40% obstruction) increased from 1.83 m/s to 3.79 m/s (+107%). Similarly, the TSS peak velocity increased from 1.83 m/s to 3.83 m/s (+109%). While the peak velocity of DSS increased from 1.83 m/s to 5.57 m/s (+204%), this had the most significant impact on velocity. Under the same percentage obstruction, the maximum speed is affected by the type of SAS; at low obstruction (10%, 20%), the TSS has the greatest impact; as the obstruction increases, the DSS peak velocity in this case is significantly higher than in the other two cases.

The velocity distribution and locations in the circulation zone did not change significantly with SAS. However, increasing the obstruction percentage forces the flow to be more lateral, affecting the pattern of eddy shedding. Figure 8b–d shows the velocity distribution on the centerline of the MHV section during the peak systole phase. When the blood passes through the SAS, the flow velocity starts to accelerate. Compared with TSS, the other two SAS tend to generate a higher peak velocity in the middle of the leaflet when the blood flow passes through the MHV.

### 3.2. Vortex Dynamic

Velocity is a measure of the speed and direction of fluid flow, while vorticity is a measure of the local rotation of fluid elements. In general, areas of high velocity tend to have a higher vorticity, while areas of low velocity tend to have a lower vorticity. In the context of blood flow simulation, the relationship between velocity and vorticity can be used to study the formation and behavior of vortices. The vorticity formation at the peak systole phase (0.2 s) is shown in Figure 9. The vortex structure’s formation depends on the type and severity of the SAS. A dominant vortex forms in the sinus area of the Valsalva behind the valve. A distinct vortex-shedding mechanism (V-shaped on the Karman vortex street) was observed, and the intensity was proportional to the percentage of SAS. In the presence of subaortic stenosis, the wake of the SAS interacts with the wake passing through the leaflets. With increasing obstruction, the vortex structure forms a complex vortex in the sinuses of Valsalva. Especially in the case of DSS, the intensity, coverage, vortex, and number of structures are significantly increased. The magnitude of the vorticity is proportional to the degree of LVOT obstruction, the vortex shedding mechanism at the trailing edge of the BMHV. Additionally, the location of the sinus and recirculation zones downstream of the valve are related to the obstruction.

Q-criterion can be used to analyze the vortex structures in the flow. The Q-criterion is calculated based on the second invariant of the velocity gradient tensor, which is a measure of the local rate of rotation in the fluid flow. High values of the Q-criterion indicate the presence of regions with strong vorticity and can be used to identify the locations of vortices in the flow. In this study, Q-criterion was used to analyze the complex flow patterns and vortical structures. These vortex structures are potentially linked to platelet activation and thrombosis. Velocity gradient, 
∇ u ∇ u= S + Ω
, where S is the rate-of-strain tensor, and Ω is the vorticity tensor:
(4)
$$S=1/2\left(\nabla u+\nabla u^{T}\right)$$


(5)
$$\Omega =1/2\left(\nabla u-\nabla u^{T}\right)$$


The vortical structure *Q* is defined as 
$Q=1/2\left({\left|\left|\Omega \right|\right|}^{2}+{\left|\left|S\right|\right|}^{2}\right)$
**S** and **Ω** denote the symmetric and antisymmetric parts of the velocity gradient, respectively, and 
$\left|\left|\cdot \right|\right|$
 is the Euclidean matrix norm. The fundamental definition of *Q* > 0 refers to those where the rotation rate dominates the strain rate and is occupied by vortical structures.

The three-dimensional vortex structure is visualized in Figure 10 using the Q-criteria for different percentages of obstruction of the three types of SAS in the peak systole phase (*t* = 0.2 s), where the colors represent the velocity value.

A small-scale chaotic vortex structure was formed before the MHV, and the structure was well organized and layered. The vortex ring disintegrated rapidly once the MHV interacted with the leaflet and rearranged into flow-oriented vortex filaments. Large vortex structures were mainly concentrated in the sinus region. In comparison to healthy MHV, due to the presence of the SAS, the three-dimensionally increased flow occurred after blood flow through the SAS. The content of the vortex structure surged; with the increase of the obstruction percentage, the large-scale transient vortices rearranged into smaller longitudinal vortex filaments that filled the entire watershed of the SAS.

Moreover, the presence of SAS made the hinge region form a more obvious vortex structure. As the speed was low, the stagnation time of blood cells may have increased. There are differences in the vortex structure of different types of SAS. TSS forms an apparent layered vortex structure on the SAS wall of the long tunnel, and DSS has the most apparent effect on the vortex with the most complex vortex structure.

### 3.3. Pressure Drop

Figure 11 shows the pressure of the cut plane along the centerline of the flow direction. In the case of a healthy MHV, the pressure in front of the anterior edge of the leaflets dropped rapidly due to MHV installation, was nearly constant through the MHV and the sinus of Valsalva, and was nearly constant in the aorta. The flow passed through the MHV and sinus of Valsalva and recovered in the ascending aorta. The presence of SAS led to a significant increase in the pressure drop along the axial direction, with the pressure drop increasing as the degree of stenosis increased. Additionally, the pressure drop was more pronounced in the vicinity of the SAS, indicating that the obstruction caused by the SAS had a significant impact on the pressure drop. This is because the obstruction of blood flow through the narrow passage of the SAS results in an increase in velocity and turbulence, which in turn leads to an increase in pressure drop. Of the three types of SAS, DSS has an obstructive surface that overlaps the aortic annulus, resulting in only a sizeable annular obstruction of the entire fluid pathway. In the case of the same obstacle percentage, the effect of DSS on the pressure drop is the most obvious. Figure 12 shows the normalized transvalvular pressure gradients (TPG). An increase in the percentage of obstruction resulted in a parabolic increase in the TPG. The TGP differences between the different types were not significant at low handicap percentages but became more pronounced between DSS and the other two SASs as the handicap percentages increased. At 40% obstruction, TSS and FSS were approximately five times higher than TPG without SAS, while DSS TPG was approximately 11 times higher than TGP without SAS and twice as high as the other two SAS. This suggests that the different types of SAS have a significant impact on the TPG.

### 3.4. Wall Shear Stress

In an MHV without SAS, the wall shear stress (WSS) on the walls and leaflets was lower during the peak phase, with an average WSS of 15.9 Pa and a maximum WSS of 223 Pa. As the blocking percentage increases, the WSS value increases in a parabolic manner. As shown in Figure 13, the largest increase in WSS was caused by SAS blocking. At the DSS type, the increase in WSS is more pronounced, and when the obstacle percentage is increased to 40%, the maximum WSS is 2.8 times that of no obstacle. The other two types of SAS, the largest WSS, had the slowest growth under the SAS barrier, at approximately 1.5 times. The average WSS displayed a similar trend. However, for FSS and DSS, at 10% and 20% obstacles, the average WSS was slightly lower than the average WSS without SAS. That’s because the SAS blocks blood flow; when passing through smaller blockages, the lower velocities and vortices that form behind the blockage create dissipation, resulting in a lower WSS on the wall behind the SAS. Ultimately, this results in a lower average WSS with blocking than MHV without blocking.

### 3.5. Doppler Parameters

Doppler echocardiography is often used as an initial evaluation method in modern diagnosis because of its availability, non-invasiveness, radiation-freeness, and cost-effectiveness. There are various Doppler quantification parameters to assess transvalvular velocity, gradient, and the Doppler velocity index (DVI) [26].

The Pressure Gradient (∆*P*) was estimated from the simplified Bernoulli equation by using the maximum transvalvular velocity (
$V_{max}$
) [27].

(6)
$$∆P=4V_{max}^{2}\;\left(\mathrm{mmHg}\right)$$


The Doppler Velocity Index (*DVI*) is the ratio of the peak velocity of the left ventricle outflow tract (LVOT) to the transvalue peak velocity [28].

(7)
$$DVI=\frac{V_{in}}{V_{max}}$$

where 
$V_{in}$
 is inlet velocity, 
$V_{max}$
 is maximum velocity within the MHV.

Aortic velocity, the most reproducible of these measures, is the strongest predictor of clinical outcome. For all cases, the peak velocity is proportional to the obstruction percentage of the SAS. When the MHV model was without SAS or mild DSS (10%), the maximum velocity did not exceed a maximum velocity amplitude of 2 m/s, which can be established as the asymptomatic threshold. However, tunnel-type SAS velocity significantly increased in the early stage of obstruction, with 10% of the obstruction already beyond the threshold, where it was (2.4 m/s). Alongside the percentage increase in obstruction, the maximum velocity exceeded the value of the asymptomatic threshold. That said, markedly different growth rates appeared between different types of SAS. It was observed that when the obstacle percentage of SAS reached 30%, the DSS peak velocity exceeded the recommended dysfunction threshold of 3 m/s. This value (3 m/s) can be used to distinguish the threshold of mild stenosis from severe stenosis. If the obstacle percentage continues to increase, the peak speed will significantly exceed this threshold.

Figure 14 displays stenosis categories: severity versus pressure gradient, proportional to the severity in all cases. The higher the severity, the greater the pressure gradient. Previous studies have found that in the absence of SAS, the pressure gradient does not exceed 20 mmHg. In this study, the pressure gradient was less than 20 mmHg when there was no or a slight obstruction (10%), and this value can be used as a potential threshold for distinguishing with or without stenosis. ACC/AHA guidelines recommend surgical intervention for peak transient echocardiographic gradients greater than 50 mmHg. When the obstacle percentage reached 30%, DSS had reached 53.8 mmHg, exceeding the threshold. But for the other two types of SAS, the threshold of 50 mmHg was only exceeded when the percentage of impairment reached 40%. Using the same threshold criteria for different types of SAS may result in delayed treatment.

The relationship between DVI and SAS is shown in Figure 15, which is inversely proportional to the severity of the stenosis (the larger the obstruction, the smaller the DVI). This threshold was reached for all types of SAS when the obstruction percentage was greater than 30%. The DVI value was greater than 0.5 in the absence of stenosis and mild stenosis, while in severe stenosis, the previous threshold of 0.3 could be used for the zone.

As can be seen from the Doppler diagnostic parameters, there are clear differences between the different types of SAS. In the case of the same diagnostic threshold for both TSS and FSS types, a higher percentage of barriers can be reached. Therefore, it is crucial to distinguish between different types of SAS. Appropriately lowering the diagnostic threshold can prevent underestimating the severity of both TSS and FSS types.

## 4. Discussion

### 4.1. Flow Characteristics in the Presence of SAS

The presence of SAS has a significant impact on the velocity, vorticity, and WSS. When blood passes through the SAS, blood flow is accelerated, and when blood passes through the MHV inlet, blood flow reaches its peak value in the middle of the leaflet. Because of the existence of the SAS, the three jets passing through the MHV will interfere with each other and enter a chaotic state ahead of time, which becomes more obvious as the degree of obstacle increases.

A significant increase in velocity will cause flow separation at the tip of the SAS and the end of the leaflet, and the vortex structures will interfere with each other to form complex eddies at the Valsalva. In particular, the DSS will lead to a significant increase in vortex intensity, coverage, eddy currents, and the number of structures. However, TSS produces obvious vorticity in its tunnel-like constriction, and an obvious layered vortex structure is formed on the SAS wall of the long tunnel. The layered vortex structure on the TSS surface would lead to an increase in WSS. Thus, at low percentage blocking (10%, 20%), the average WSS of TSS is higher than that of the other two. As the severity of SAS increases, the velocity of the wake at the end of the SAS significantly increases and affects the leaflet. Accordingly, more complex and stronger eddies lead to a sharp increase in WSS. The high-speed jet impact and the increased WSS will increase the shear force on the blood. Due to blood damage, the possibility of platelet activation will increase, and the complex vortex in the sinus area may increase the stagnation time in the recirculation area, which may cause platelet accumulation leading to thrombosis. Thrombus formation around the MHV hinge region may contribute to MHV dysfunction.

### 4.2. Differentiation of Severity of Subaortic Stenosis

The severity of subaortic stenosis is particularly important in patient diagnosis. The main parameters to assess the severity of SAS include peak velocity (
$V_{peak}$
), TPG. It can be measured directly with continuous-wave Doppler.

The classification of SAS severity grades is shown in Table 1. Generally, these parameters should be consistent. A peak velocity that is greater than 
4 m/s
 and TPG that is greater than 80 mmHg are defined as severe SAS. However, some studies suggest that resection should only be considered in certain circumstances [29,30]. However, the general consensus reported in the 2008 ACC/AHA guidelines recommends surgical intervention for patients with transient echocardiographic peak gradients greater than 50 mmHg [31]. That is to say, moderate SAS can be treated surgically. For patients with implanted MHV, early treatment is even more important because the risk of MHV failure increases with the severity of SAS. For TSS and FSS, the threshold of moderate risk was reached at higher barrier percentages.

### 4.3. Limitations

This study has several limitations that should be addressed in future research. First, the study assumes SAS to be a rigid structure and does not consider the material properties of SAS itself. In particular, for the study of DSS, the fluid-structure interaction should be used to set the SAS as elastic to capture the deformation caused by fluid erosion and improve the accuracy of the results.

Additionally, this study focuses on the systolic phase of the cardiac cycle, and the MHV leaflet is fully open at this time, which does not involve the entire cardiac cycle. Therefore, to study the influence of SAS on leaflet movement, the entire cardiac cycle should be considered.

To address these limitations, in vitro experiments can be incorporated into the research methodology to validate the CFD results and obtain more accurate data. A more accurate three-dimensional (3D) structure of the SAS can be obtained through CT scans or other imaging techniques to create a more detailed and realistic model of the SAS for the CFD simulations. Simulating the SAS and blood vessels using elastic materials can help in modeling the dynamic behavior of the SAS and blood flow more accurately.

Furthermore, collaborating with experts in the field can help in incorporating more advanced CFD techniques, such as particle image velocimetry (PIV) or computational particle tracking (CPT), to obtain more accurate and detailed information about the flow patterns and turbulence within the SAS and blood vessels.

Overall, addressing these limitations, incorporating in vitro experiments, obtaining a more accurate 3D structure of the SAS, simulating tissues with elastic materials, and using advanced CFD techniques can all help in overcoming the limitations of this thesis and obtaining more accurate and detailed results.

## 5. Conclusions

In the present study, the flow patterns of subaortic stenosis in MHV in an unsteady state were investigated. The presence of SAS significantly increases velocity, pressure drop, vorticity, and WSS. In the presence of SAS, a high-velocity jet is formed to flush the MHV leaflet and the wall surface, resulting in a higher WSS and increasing the chance of platelet activation or hemolysis. Among the different types of SAS, studies have shown that DSS is significantly different from the other two and tends to produce higher parameters in the presence of small proportions of obstruction, so the risk in the presence of DSS is greater than that of TSS and FSS. The TSS produces a separated vortex structure on its tunnel-like, narrow surface, resulting in a higher WSS at a low degree of hindrance. The presence of the SAS results in the formation of high-velocity jets and complex eddies before the blood flow passes through the MHV. As the severity of SAS increases, it will increase the possibility of MHV failure and cause a series of complications. Therefore, early treatment is especially important for patients with implanted MHV. The study also highlights the impact of SAS types on simulation results and diagnostics. Finally, further studies on irregular stenosis and its relative MHV need to be performed to investigate the impact of SAS on MHV in greater depth.

## Figures and Tables

**Figure 1 bioengineering-10-00312-f001:**
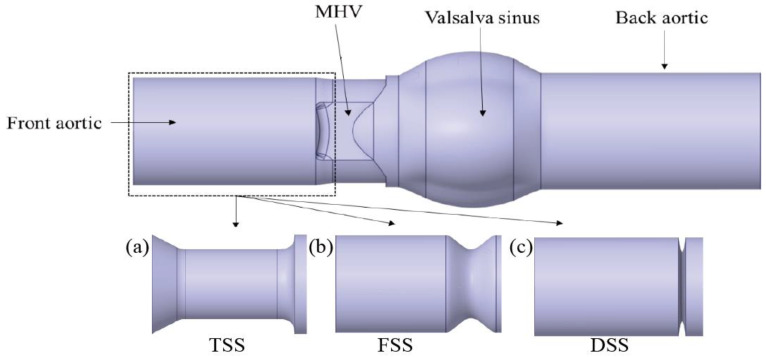
MHV structure and obstruction complications ((**a**) TSS, (**b**) FSS, (**c**) DSS).

**Figure 2 bioengineering-10-00312-f002:**
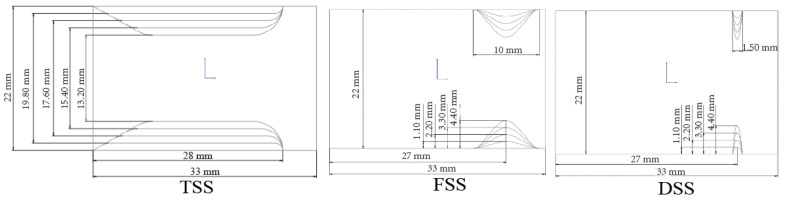
Model geometry with dimensions for three types of SAS (obstruction height: 1.1, 2.2, 3.3, and 4.4 mm).

**Figure 3 bioengineering-10-00312-f003:**
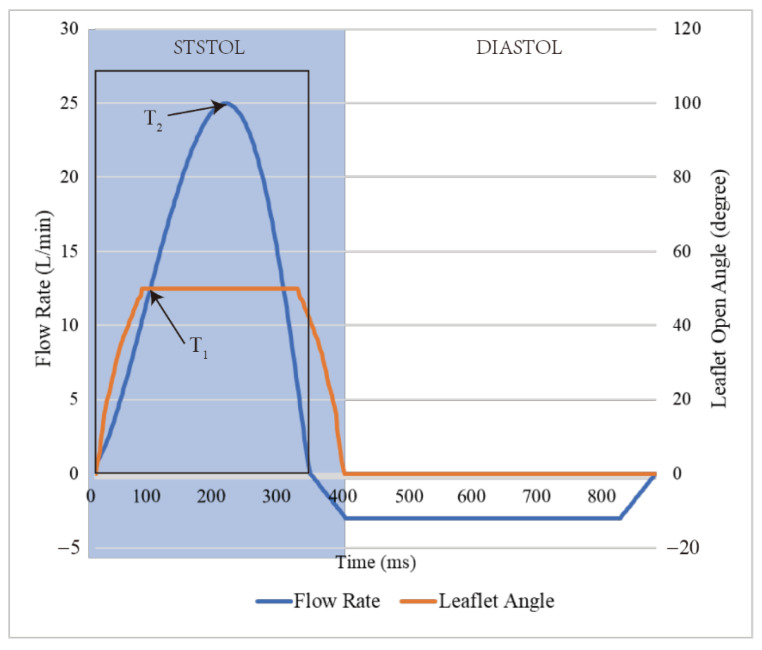
Flow rate pulse of the aorta inlet and leaflet open angle changes.

**Figure 4 bioengineering-10-00312-f004:**
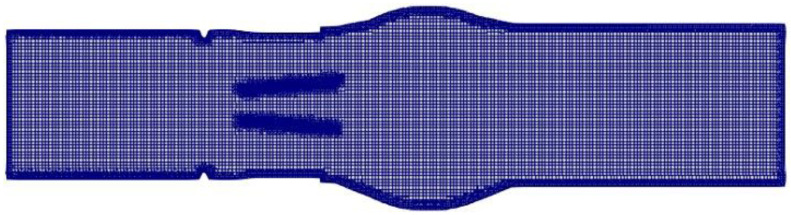
Mesh quality of the MHV model (Center Section View).

**Figure 5 bioengineering-10-00312-f005:**
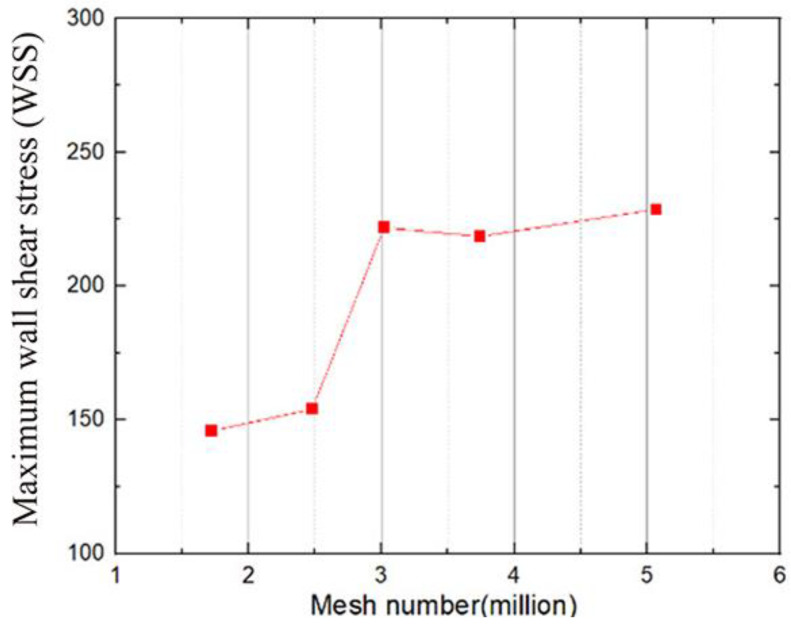
Mesh independence verification.

**Figure 6 bioengineering-10-00312-f006:**
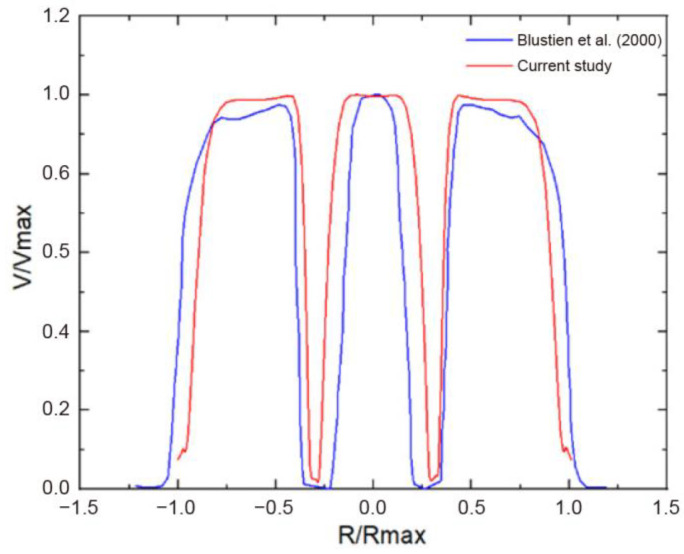
Comparison between the normalized velocities of the current study and the numerical study of Blustien et al. [25].

**Figure 7 bioengineering-10-00312-f007:**
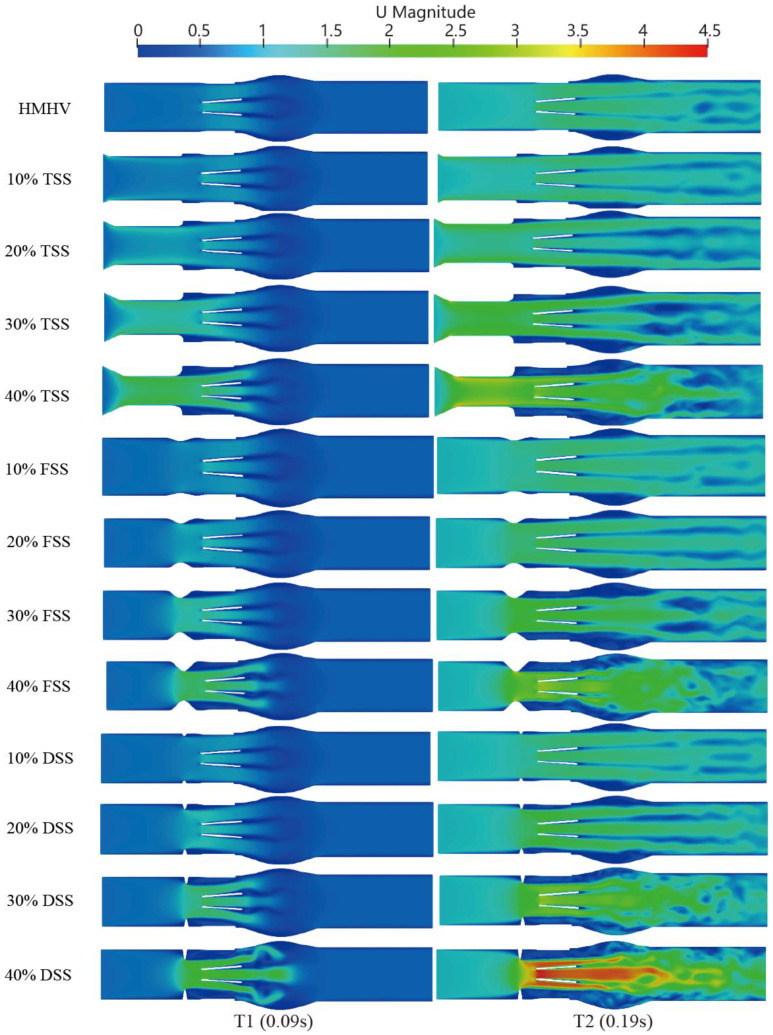
Velocity contours at SAS1, SAS2, and SAS3 in different percentage blockage (10%, 20%, 30%, and 40%) at T1 (peak) and T2 (deceleration phase).

**Figure 8 bioengineering-10-00312-f008:**
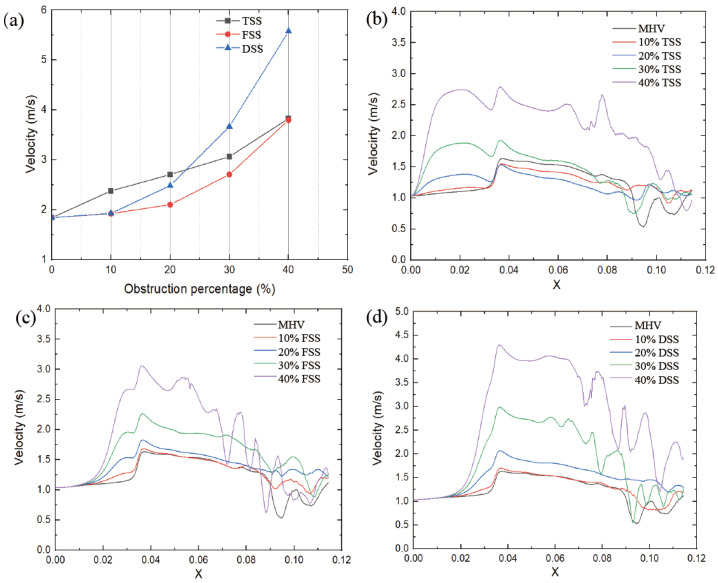
Velocity comparison of (**a**) obstruction vs. peak velocity and velocity change along the centerline (**b**) TSS, (**c**) FSS, (**d**) DSS (*t* = 0.2 s).

**Figure 9 bioengineering-10-00312-f009:**
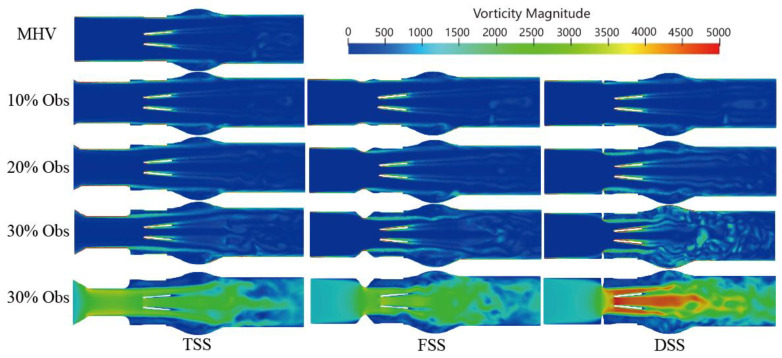
Vorticity distributions downstream without and with SAS at peak time (*t* = 0.2 s).

**Figure 10 bioengineering-10-00312-f010:**
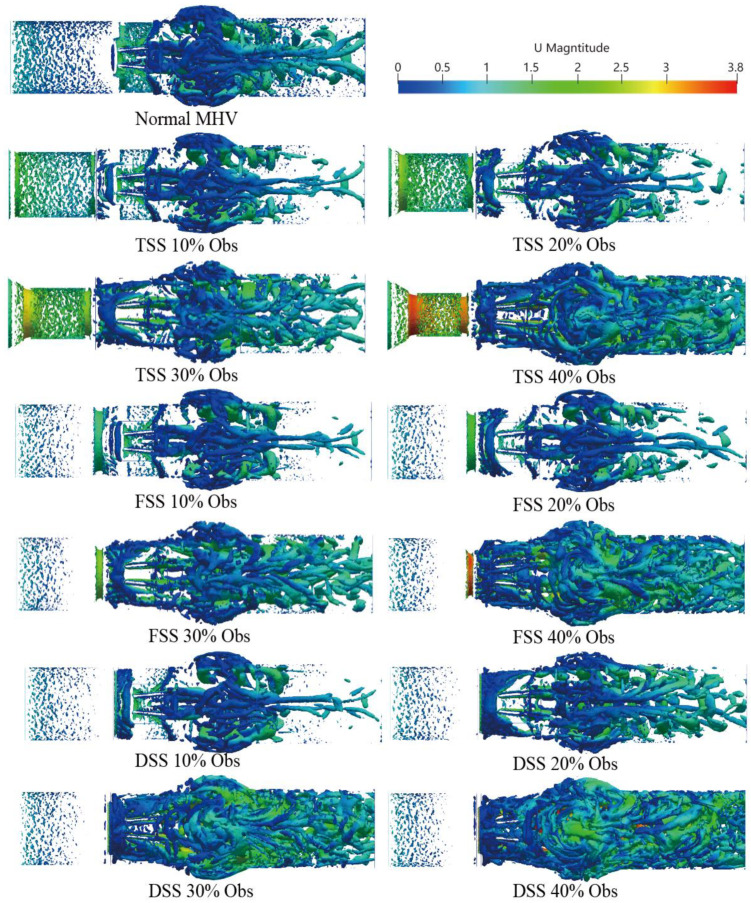
Vortical structure determined by the Q-criterion (Q = 10,000), colored by velocity magnitude, is obtained in three types of SAS in the peak flow phase (*t* = 0.2 s).

**Figure 11 bioengineering-10-00312-f011:**
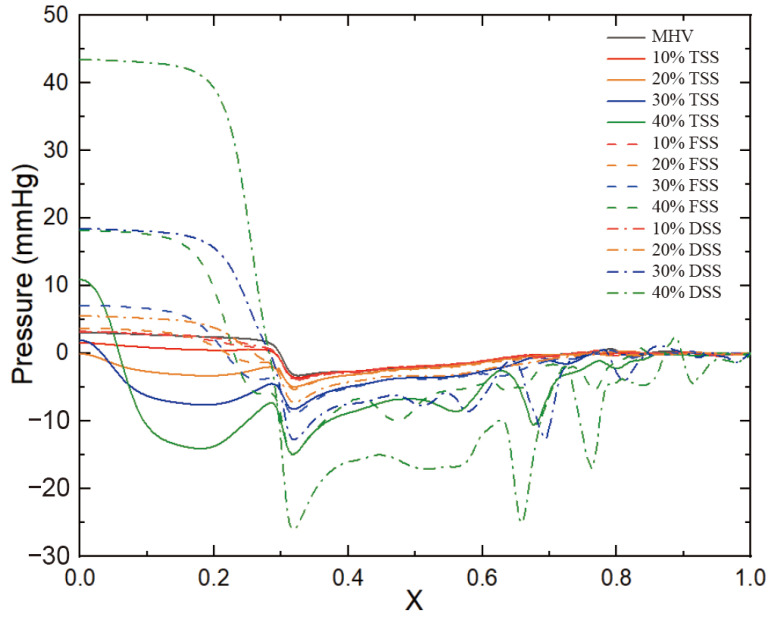
Pressure drops in the direction of blood flow (*t* = 0.2 s).

**Figure 12 bioengineering-10-00312-f012:**
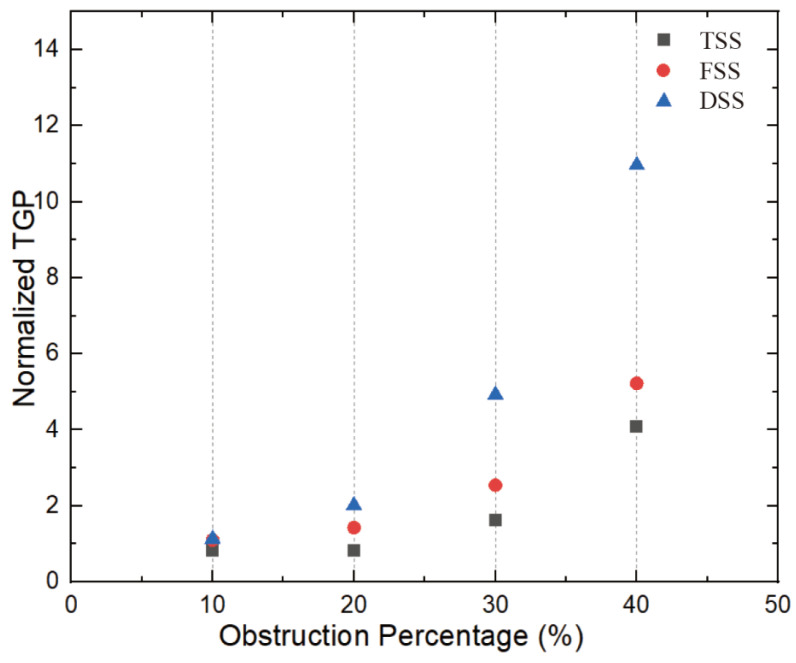
NTGP VS obstruction percentage (*t* = 0.2 s).

**Figure 13 bioengineering-10-00312-f013:**
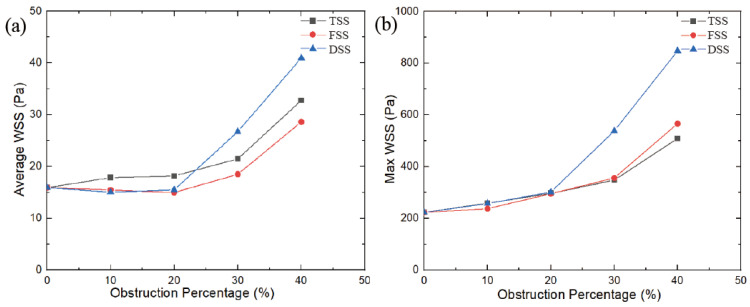
WSS VS obstruction percentage (**a**) Average WSS, (**b**) Max WSS (*t* = 0.2 s).

**Figure 14 bioengineering-10-00312-f014:**
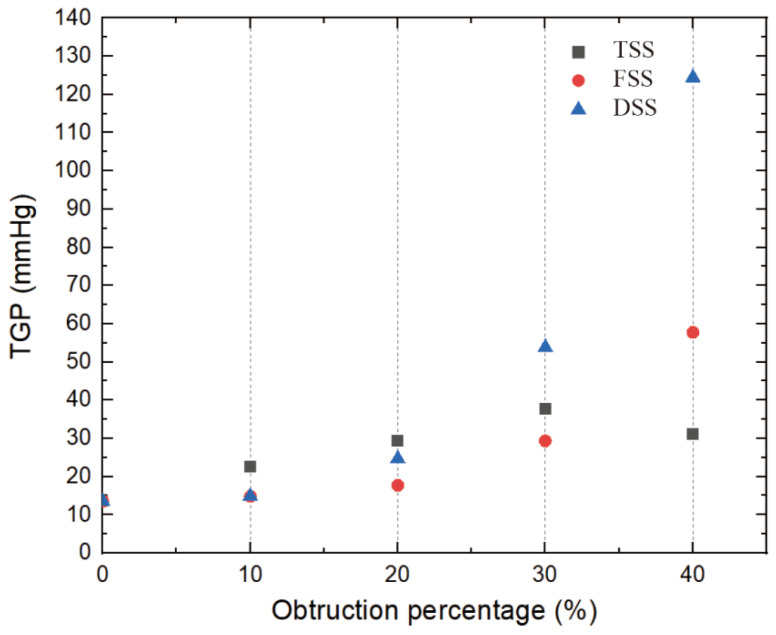
Mean transvalvular pressure gradient for three types of SAS at different obstruction percentages (*t* = 0.2 s).

**Figure 15 bioengineering-10-00312-f015:**
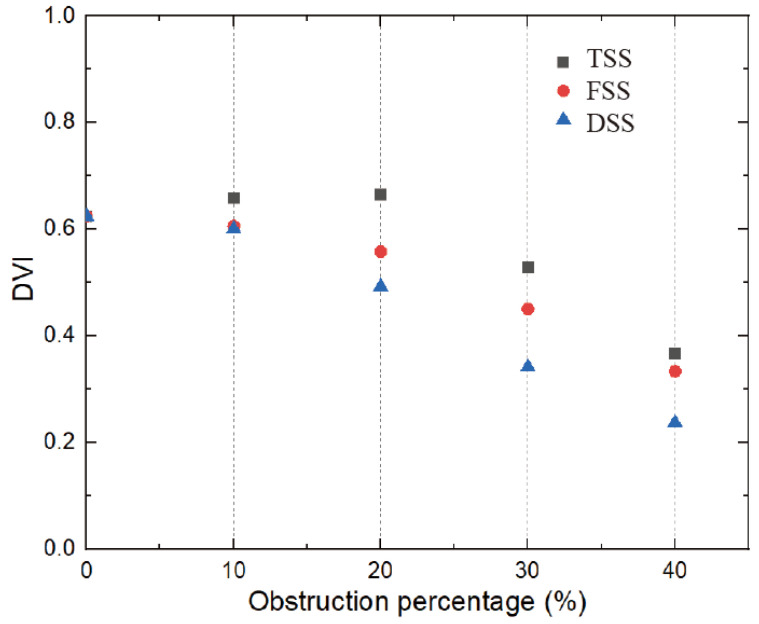
The relationship between obstruction percentage and Doppler velocity index (*t* = 0.2 s).

**Table 1 bioengineering-10-00312-t001:** Severity grade of subaortic stenosis.

Severity Grade	TPG [32]	$V_{peak}$ [33]
Normal	<20 mmHg	<2.5 m/s
Mild SAS	20–50 mmHg	2.5–3m/s
Moderate SAS	50–80 mmHg	3–4 m/s
Severe SAS	>80 mmHg	>4 m/s

## Data Availability

Data are contained within the article.

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
