# Peer review of "Hemodynamic Effects of Subaortic Stenosis on Blood Flow Characteristics of a Mechanical Heart Valve Based on OpenFOAM Simulation"

_bioengineering, 2023, doi:10.3390/bioengineering10030312_

Round 1
Reviewer 1 Report (Previous Reviewer 1)
The abstract is still not clear enough and should be improved.
Author Response
The abstract is still not clear enough and should be improved.
Replay: modified abstract part
Subaortic stenosis (SAS) is a common congenital heart disease that can cause significant morbidity and mortality if not treated promptly. Patients with heart valve disease are prone to complications after replacement surgery, and the existence of SAS can accelerate disease progression, so timely diagnosis and treatment are required. However, the effects of subaortic stenosis on mechanical heart valves (MHV) are unknown. This study aimed to investigate flow characteristics in the presence of subaortic stenosis and computation-ally quantify the effects on the hemodynamics of MHV. Through the numerical simulation method, the flow characteristics and related parameters in the presence of SAS can be more intuitively observed. Based on its structure, there are three types of SAS: Tunnel-type SAS (TSS); Fibromuscular annulus SAS (FSS); Discrete SAS (DSS). The first numerical simulation study on different types of SAS found that there are obvious differences among different types of SAS. Among them, the tunnel-type SAS formed a separated vortex structure on the tunnel-type narrow surface, which exhibits higher wall shear force at a low obstacle percentage. However, discrete SAS showed obvious differences when there was a high percentage of obstacles, forming high peak flow, high wall shear stress, and high-intensity complex vortex. The presence of all three types of SAS results in the formation of high velocity jets and complex vortices in front of the MHV, leading to increased shear stress and stagnation time. These hemodynamic changes significantly increase the risk of MHV dysfunction and the development of complications. Despite differences be-tween the three types of SAS, the resultant effects on MHV hemodynamics are consistent. Therefore, early surgical intervention is warranted in SAS patients with implanted MHV.
Reviewer 2 Report (New Reviewer)
I have read the manuscript “Hemodynamic Effects of Subaortic Stenosis on the Blood Flow Characteristics of Mechanical Heart Valves” submitted to Bioengineering, MDPI.
The subject of this manuscript is interesting and the authors assembled valuable information on the influence of different the effects of subaortic stenosis on mechanical heart valves, using simulation. The document is comprehensive, the discussion is reasonable, however; some corrections need to be made for the current document. So, I think this manuscript should be considered for publication in Bioengineering, MDPI only after minor proper modifications. Some of my specific comments are below:
Point 1: I consider that the authors could slightly modify the title of the manuscript to indicate the tool or approach they use to carry out their work. Simulation by computationally modeling can be added to title.
Point 2: In the abstract, the authors must make clear to the readership what are the types of SAS introduced and compared, for example they mention “Based on its structure, there are three types of SAS.” In the introduction can be observed which these structures are, however it should be mentioned in an abbreviated way in the abstract for a better lector understand.
Point 3: In the abstract, the authors should mention if it is the first that the SAS behavior differences are reported and if this is only possible to carry out by simulation This finding seems relevant.
Point 4: In the abstract, the authors include the specific importance for one SAS-type, for example they mentioned “SAS formed high-speed jets … which caused the blood flow to stand higher shear stress on the MHV”. Please, include the characteristics/implications that the other SAS represent for the MHV.
Point 5: Material and methods section, for the sentence Page 4, line 125 “The simulation was carried out under pulsatile conditions… and a 0.86 s cardiac cycle. Suitable reference should be included. Page 3, line 116, typo for 28mm.
Point 6: Material and methods section, for the sentence Page 4, line 125 “This study ignores leaflet opening and closing phases and mainly studies the valve full opening phase”. Additional explication should be added.
Point 7: Material and methods section, for the sentence. Please indicate why the density value was set at 1060 kg⁄m3, indicate clearly the liquid used for the experiments and what are its characteristics.
Point 8. Material and methods section, indicate the meaning for the parameters of equation 3.
Point 9. Figure 6 Please, change axis title of WSS to maximum wall shear stress (WSS).
Point 10. Figure 7 Please, modify this figure for a better visualization of the titles on its axes
Point 11. It is possible for the authors to provide an estimate or idea of the accuracy of the results obtained taking into account the mentioned limitations of the present study?
Author Response
Point 1: I consider that the authors could slightly modify the title of the manuscript to indicate the tool or approach they use to carry out their work. Simulation by computationally modeling can be added to title.
Replay: Change “Hemodynamic Effects of Subaortic Stenosis on the Blood Flow Characteristics of Mechanical Heart Valves” to “Hemodynamic effects of subaortic stenosis on blood flow characteristics of mechanical heart valve based on OpenFOAM simulation”
Point 2: In the abstract, the authors must make clear to the readership what are the types of SAS introduced and compared, for example they mention “Based on its structure, there are three types of SAS.” In the introduction can be observed which these structures are, however it should be mentioned in an abbreviated way in the abstract for a better lector understand.
Replay: Based on its structure, there are three types of SAS: Tunnel-type SAS (TSS); Fibromuscular annulus SAS (FSS); Discrete SAS (DSS).
Point 3: In the abstract, the authors should mention if it is the first that the SAS behavior differences are reported and if this is only possible to carry out by simulation This finding seems relevant.
Replay: Through the numerical simulation method, the flow characteristics and related parameters in the presence of SAS can be more intuitively observed. Based on its structure, there are three types of SAS: Tunnel-type SAS (TSS); Fibromuscular annulus SAS (FSS); Discrete SAS (DSS). The first numerical simulation study on different types of SAS found that there are obvious differences among different types of SAS.
Point 4: In the abstract, the authors include the specific importance for one SAS-type, for example they mentioned “SAS formed high-speed jets … which caused the blood flow to stand higher shear stress on the MHV”. Please, include the characteristics/implications that the other SAS represent for the MHV.
Replay: The presence of all three types of SAS results in the formation of high velocity jets and complex vortices in front of the MHV, leading to increased shear stress and stagnation time. These hemodynamic changes significantly increase the risk of MHV dysfunction and the development of complications. Despite differences between the three types of SAS, the resultant effects on MHV hemodynamics are consistent.
Point 5: Material and methods section, for the sentence Page 4, line 125 “The simulation was carried out under pulsatile conditions… and a 0.86 s cardiac cycle. Suitable reference should be included. Page 3, line 116, typo for 28mm.
Replay: The simulation was carried out under pulsatile conditions, utilizing an experimental pulsatile flow as the inlet condition and applying typical physical flow conditions of approximately 25 L/min peak flow rate and a 0.86 s cardiac cycle [13]
Point 6: Material and methods section, for the sentence Page 4, line 125 “This study ignores leaflet opening and closing phases and mainly studies the valve full opening phase”. Additional explication should be added.
Replay: present study is focused on investigating the hemodynamic behavior of the mechanical heart valve during the full opening phase from 45 to 280 ms,, and thus it does not account for leaflet opening and closing phases. Specifically, the study aims to investigate the interaction between blood flow and valve leaflets during the peak flow phase, while also analyzing the formation of complex eddies and turbulence, and the distribution of shear stress on the valve surface.
Point 7: Material and methods section, for the sentence. Please indicate why the density value was set at 1060 kg⁄m3, indicate clearly the liquid used for the experiments and what are its characteristics.
Replay: Blood density set at 1060 kg⁄m3 is depending on the reference.[24]
Point 8. Material and methods section, indicate the meaning for the parameters of equation 3.
Replay: is the viscosity at zero shear rate, is the viscosity at infinite shear rate, is a time constant of the fluid, is the shear rate, n is a dimensionless parameter that characterizes the degree of shear-thinning.
Point 9. Figure 6 Please, change axis title of WSS to maximum wall shear stress (WSS).
Replay:
Point 10. Figure 7 Please, modify this figure for a better visualization of the titles on its axes
Replay:
Point 11. It is possible for the authors to provide an estimate or idea of the accuracy of the results obtained taking into account the mentioned limitations of the present study?
Replay: This study has several limitations that should be addressed in future research. First, the study assumes SAS to be a rigid structure and does not consider the material properties of SAS itself. In particular, for the study of DSS, the fluid-structure interaction should be used to set the SAS as elastic to capture the deformation caused by fluid erosion and improve the accuracy of the results.
Additionally, this study focuses on the systolic phase of the cardiac cycle, and the MHV leaflet is fully open at this time, which does not involve the entire cardiac cycle. Therefore, to study the influence of SAS on leaflet movement, the entire cardiac cycle should be considered.
To address these limitations, in vitro experiments can be incorporated into the research methodology to validate the CFD results and obtain more accurate data. A more accurate three-dimensional (3D) structure of the SAS can be obtained through CT scans or other imaging techniques to create a more detailed and realistic model of the SAS for the CFD simulations. Simulating the SAS and blood vessels using elastic materials can help in modeling the dynamic behavior of the SAS and blood flow more accurately.
Furthermore, collaborating with experts in the field can help in incorporating more advanced CFD techniques, such as particle image velocimetry (PIV) or computational particle tracking (CPT), to obtain more accurate and detailed information about the flow patterns and turbulence within the SAS and blood vessels.
Overall, by addressing these limitations, incorporating in vitro experiments, obtaining a more accurate 3D structure of the SAS, simulating tissues with elastic materials, and using advanced CFD techniques can all help in overcoming the limitations of this thesis and obtaining more accurate and detailed results.
Reviewer 3 Report (New Reviewer)
Overall, the manuscript needs to be improved in terms of the methodology and experimental results involved. The results should be more interesting if the authors provide a connection between the section. Figure 1 has been mentioned that it is taken from Ref[20] while figure 2 from [21], can the authors explain why this problem is so important to take from [20] and [21]?
Figures’ captions should be informative. Section 2.2 , ‘fluid boundary conditions were obtained from prior numerical simulations of the flow through the MHV in an aortic setting [13]’, it should be better that the authors should explain in the last paragraph of the introduction the aims of this work.
What is T1 and T2? It needs to be defined.
Provide a reference for equations (1) and (2)
The constant for density should be checked.
It seems that the authors used the previous results for results compilation, and the only thing which is new is the simulation tool box. The authors need to provide some solid results for establishing the results.
There is nothing important in the simulations that can convince the readers, therefore some useful results must be supplied.
The graphical results and their captions must be explained in more details.
Author Response
Point1. Figure 1 has been mentioned that it is taken from Ref[20] while figure 2 from [21], can the authors explain why this problem is so important to take from [20] and [21]?
Replay: Ref[20] fully reviewed SAS and firstly described three different types SAS, and give the simplified SAS structure figure. In this study mainly concentrate on compare the difference between three different type of SAS, so the first step is generate SAS model, so the general structure is generate from Ref [20]. And in the Ref [21] descripted the structure size details, like the SAS root width range. So in figure 2, the size details is draw based on ref [21]
Point 2. Figures’ captions should be informative. Section 2.2 , ‘fluid boundary conditions were obtained from prior numerical simulations of the flow through the MHV in an aortic setting [13]’, it should be better that the authors should explain in the last paragraph of the introduction the aims of this work.
Point 3. What is T1 and T2? It needs to be defined.
= 0.09 s at the mid-acceleration phase and = 0.19 s at peak systole. These are two important time nodes in the simulation, and the subsequent result analysis will mainly focus on these two time points
Point 4. Provide a reference for equations (1) and (2)
Added reference [23]
Point 5. The constant for density should be checked.
The density is set depending on reference [24]“Marom, G. and Bluestein, D., 2016. Lagrangian methods for blood damage estimation in cardiovascular devices-How numerical implementation affects the results. Expert review of medical devices, 13(2), pp.113-122.”
Point 5: The graphical results and their captions must be explained in more details.
Replay: Modified in the manuscript
Round 2
Reviewer 3 Report (New Reviewer)
Accepted
This manuscript is a resubmission of an earlier submission. The following is a list of the peer review reports and author responses from that submission.
Round 1
Reviewer 1 Report
Chen et al assessed a model of different types of subaortic stenosis on mechanical prosthetic valve flow /
The study is interesting however there are some points that need to be verified
1) The title is somewhat misleading :
Hemodynamic effects of prosthetic heart valves in subaortic stenosis-it is more the effect of Subaortic stenosis on prosthetic valve
2) Abstract: SAS3 –Should be defined or described since using the group name without explanation is not clear when you read the abstract
3) The definitions of SAS 1.2 3 is not clear in lines 119-122. The figure also shows 2 types. The definition is not the same as the order of SAS types in introduction lines 36-43 and it is confusing
4) "Although fluoroscopy or computed tomography can assess the severity of mechanical heart valves (MHV) and SAS [2],-what is the meaning of severity of mechanical valve??
5) The authors' mention several times effective orifice area (EOA) of SAS-, the Bernoulli equation is usually used for AVA without SAS.
6) In lines 327-337, the authors say that with increased SAS there is a gradient that is above the 20mmHG threshold of normal mechanical valve. However the gradient are not due to valve malfunction, they are due to SAS
7) In the presence of SAS, a high-velocity jet will be formed to flush the MHV and the wall surface, resulting in a higher WSS, increasing the chance of platelet activation or hemolysis, and promoting the rapid growth of SAS and the formation of pannus. The authors showed increase in WSS that potentially cause platelet activation and inflammation. The hemolysis is possible ' however not shown. The authors should explain why this will promote rapid growth of SAS ? Is pannus also a type of SAS?
Reviewer 2 Report
Authors aimed": to evaluate current Doppler parameters for the accurate and early detection of mechanical valve dysfunction and computationally quantify the impact of subaortic stenosis on mechanical heart valve hemodynamics". From a strict clinical point of view, this theoretically challenging scenario might only result as a cause of misdiagnosis (both at preoperative and intraoperative evaluation). The complex computational model is impressive, nevertheless, its possible applications in a clinical setting are hard to imagine and need a clearer explanation. The manuscript needs language editing. The introduction should be completely reworked for the sake of clarity. Indeed, it seem the juxtaposition of three or four different paragraphs and does not properly convey the background and the need for the present study. Discussion should focus on the clinical significance of the complex computational results, and, again trying to make clear the possible scenarios in which such speculations might induce new treatment algorithms. A limitation paragraph should also be developed. All in all, the manuscript needs extensive rework .